# Demystifying Chronic Kidney Disease of Unknown Etiology (CKDu): Computational Interaction Analysis of Pesticides and Metabolites with Vital Renal Enzymes

**DOI:** 10.3390/biom11020261

**Published:** 2021-02-10

**Authors:** Harindu Rajapaksha, Dinesh R. Pandithavidana, Jayangika N. Dahanayake

**Affiliations:** Department of Chemistry, Faculty of Science, University of Kelaniya, Dalugama, Kelaniya 11600, Western Province, Sri Lanka; harindurajapaksha@gmail.com (H.R.); dinesh@kln.ac.lk (D.R.P.)

**Keywords:** CKDu, renal enzymes, pesticides, metabolites, molecular docking, molecular dynamics

## Abstract

Chronic kidney disease of unknown etiology (CKDu) has been recognized as a global non-communicable health issue. There are many proposed risk factors for CKDu and the exact reason is yet to be discovered. Understanding the inhibition or manipulation of vital renal enzymes by pesticides can play a key role in understanding the link between CKDu and pesticides. Even though it is very important to take metabolites into account when investigating the relationship between CKDu and pesticides, there is a lack of insight regarding the effects of pesticide metabolites towards CKDu. In this study, a computational approach was used to study the effects of pesticide metabolites on CKDu. Further, interactions of selected pesticides and their metabolites with renal enzymes were studied using molecular docking and molecular dynamics simulation studies. It was evident that some pesticides and metabolites have affinity to bind at the active site or at regulatory sites of considered renal enzymes. Another important discovery was the potential of some metabolites to have higher binding interactions with considered renal enzymes compared to the parent pesticides. These findings raise the question of whether pesticide metabolites may be a main risk factor towards CKDu.

## 1. Introduction

Chronic kidney disease (CKD) is a global health and economic issue. It is characterized by a gradual decrement of glomerular filtration rate (GFR) over time due to structural and functional defects of kidney in urinalysis, biopsy, and imaging [1]. The leading causes for CKD are thought to be glomerulonephritis, hypertension, and diabetes mellitus [2]. CKDu is an evolving health problem in some low and middle-income nations, such as El Salvador, Egypt, Cuba, Sri Lanka, Bangladesh, and India [3]. The poor rural areas are most affected with agricultural work being the dominant occupation. Poverty with the absence of access to health care makes it hard to determine the clinical features of CKDu. In Nicaragua, Central America, the highest prevalence of CKDu has been reported with 10–20% cases among the adult population [4], where the third to fifth decade age group is highly affected. In Southern India and Sri Lanka, CKDu prevalence is 1.6 and 1.5%, respectively (ref), where a wide age range is affected. Generally, the male prevalence is significantly higher than in females [5,6].

Progression of CKDu is usually symptomless until advanced phases of the disease, in which the kidneys are irreversibly damaged, resulting in mortality unless dialysis or transplantation occurs [7]. The CKDu endemic in Central America is called Mesoamerica Nephropathy (MeN). The patients have elevated serum creatinine levels and normal albuminuria [8]. Podocytic changes, Glomerulosclerosis and moderate tubulointerstitial damages were observed in MeN patient kidneys [9]. This suggests that the added stress on kidneys due to loss of water from the body from sweating increases the risk factor for CKDu [10].

Given its unknown origin, CKDu has spurred a variety of investigative efforts in recent years. A number of research studies have been completed to identify the risk factors. However, up to the present date, no specific reason has been proven scientifically to be the exact cause. There are many factors thought to increase the getting CKDu, such as dry climate, genetic vulnerability and family history of CKD, the hardness of water, and occupational exposure to Agrochemical [11]. There are many proposed courses for CKDu, such as pesticides, inorganic fertilizers, heavy metals, fluoride and hardness of drinking water, and cyanobacterial toxins [12].

In the present study, the focus is on the studying effects of pesticides and their metabolites towards CKDu. The exact mechanism of CKDu due to pesticides is unknown. It is that the increased oxidative stress [13,14,15] due to inhibition or manipulation of enzymes, such as Cytochrome P450 (CP450) and Glutathione S Transferase (GST), by pesticides, and their metabolism plays a role [16,17]. Understanding inhibition or manipulation of vital renal enzymes, such as Adenosine monophosphate (AMP) Activated Protein Kinase (AMPK), Protein Kinase C (PKC), Glutaminase (GLS), Apoptosis Signaling Kinase 1 (ASK1), and Acetylcholinesterase (AChE), by pesticide and metabolites may be a key to understand the link between CKDu and pesticides.

It is suggested that interactions of different potential agents cause CKDu rather than a single causative agent; thus, CKDu has also been termed as CKD of multi-factorial origin (CKD-mfo) [18]. These factors include the negative effects of overuse of agrochemicals, the effect of heavy metals, and other environmental pollutants which are synthetic compounds used in industrial, agricultural, and domestic use and known as xenobiotic agents, present in the environment. The potential interactions and synergism between probable agents have not been studied thoroughly. Pesticides are xenobiotic compounds. There are mainly 2 classes of enzymes that are involved in xenobiotic biotransformation, phase I enzymes and phase II enzymes [19]. CP450 has a role as phase I enzymes by increasing the hydrophilicity or nucleophilicity which aid in the elimination process and to make suitable substrate for phase II enzymes. GST has a role as phase II enzyme, which are responsible for the conjugate formation [15,17].

Rajani R et al. [20] have reported that the activation of AMPK protein slows down the progression of CKD. The effect of AMPK activation on kidney disease states is a reduction in epithelial-mesenchymal transdifferentiation, apoptosis, fibrosis, cyst formation, metabolic memory, inflammation, and cell grown and proliferation. AMPK activation also increases autophagy. The initial observations in CKDu patients are that the fibrosis of interstitial tubules, the defective function of AMPK, can be a reason for that.

When it comes to the PKC protein, balanced action of PKC protein is needed for proper renal function. How unbalanced PKC can cause renal damage and CKD can be summarized as follows. PKC participates in cellular signal transduction pathways, proliferation, differentiation, cell cycle, and apoptosis [21]. These cellular activities are linked to tumor development and cell proliferation. PKC-dependent NAD(P)H oxidase activation and alteration of mitochondrial metabolism leads to an increase of reactive oxygen species (ROS) in the cellular environment. ROS plays a central role in the excessive extracellular matrix (ECM) synthesis and degradation in the glomeruli and tubulointerstitium, leading to renal fibrosis [22]. Excessive ROS can activate a signal transduction cascade involving the mitogen-activated protein kinase and the Janus kinase/signal transducers and activators of transcription. There is then subsequent upregulation of the pro-fibrotic cytokine transforming growth factor-beta1, along with other pro-fibrotic factors, such as angiotensin II, that cause further fibrosis and ECM synthesis [23].

Glutaminase catalyzes the first step in the metabolism of glutamine by converting glutamine to glutamate. Soomro I et al. [24] analyzed the hypothesis that glutamine metabolism also plays a critical role in cyst formation and found a positive link between cyst formation and glutaminase activity.

ASK1 is a member of the mitogen-activated protein kinase kinase kinase (MAPKKK) family that activates c-Jun N-terminal kinase (JNK) and MAPK p38 in reaction to a variety of stress disorders, such as oxidative stress, endoplasmic reticulum stress, and calcium inflow. Activation of JNK signaling is a prevalent characteristic in most types of human kidney injury, both in glomerular and tubular cells, as well as in leukocyte infiltration [25]. MAPKs regulate a broad range of cellular functions. Defective activation or activity of ASK1 will lead to the development of CKD. In experimental models of acute and chronic renal disease, activation of ASK1 and subsequent activation of downstream APKs induce inflammation, fibrosis, and apoptosis and are associated with human kidney disease [26]. Downstream P38 and JNK elevated activities lead to apoptosis, necrosis, inflammation, and fibrosis in renal vasculature, glomerulus, and tubulointerstitium kidneys in diabetic and non-diabetic individuals, leading to CKD [27]. Fibrosis is a classic observation in CKDu kidneys [9].

AChE is an extracellular glycoprotein, which can significantly alter renal function changing neural, humoral, and metabolic activities. Organophosphate pesticides inhibit AChE, and AChE activity in red blood cells was shown to be significantly lower in AChE inhibitor pesticide exposed-CKD patients as compared to unexposed-CKDu patients [28]. Yeato G et al. [29] conducted a study on the level of AChE activity on Chronic Renal Failure (CRF) patients before and after dialysis. They considered the effect of aging on the AChE levels. Yeato G et al. states that AChE activity of CRF patients was significantly higher than that of the control [29].

Depending on the xenobiotic compound, in this case, pesticides, wide variety of metabolites can be formed. These formed metabolites can be seen in blood, as well as in urea, of pesticide exposed individuals [30,31], and they can be more toxic than the original pesticide itself. During the metabolism activation of the pesticide happens, this may causes the increase in toxicity this is called bio activation [32]. A prime example of bioactivation is the biotransformation of dichlorodiphenyltrichloroethane (DDT—which is not highly toxic to birds), into dichlorodiphenyldichloroethylene (DDE), which causes thinning of eggshells because it disrupts calcium metabolism [33]. Costa LG [34] reported that the metabolites formed during the metabolisms of organophosphate (OP) are more toxic. For example, oxons are produced during the metabolisms of OP and it can bind to cholinesterase or undergo hydrolysis to a dialkyl phosphate and a hydrolyzed organic moiety specific to the pesticide [35]. Cloyd [36] reported that the metabolites of chlorfenapyr, indoxacarb, malathion, and imidacloprid are active, and these metabolites may have a different mode of action when compared with the original pesticide. Sandrini et al. [37] found that metabolites of glyphosate inhibiting activity. In the case of chlorpyrifos metabolite, oxon is more toxic chlorpyrifos itself. Ethylenethiourea is the metabolite formed form dithiocarbamates, and it has been proven that ethylenethiourea can induce thyroid cancer and modify thyroid hormones [38]. Amorós et al. [39] reported that the metabolites of Fenitrothion are more toxic than the original compound. Quinalphos metabolite 2-hydroxyquinoline (HQ) has been shown to photo catalytically destroy antioxidant vitamins and biogenic amines in vitro. Riediger et al. [40] investigated the toxicity, cellular stress, and mutagenicity of HQ and found it causes oxidative damage and mutations. DDT and its metabolites has an endocrine disrupting effect, and this is more extreme in metabolites [41].

There is a lack of insight regarding the toxicity of pesticide metabolites. It is very important to take metabolites into account when investigating the relationship between CKDu and pesticides. A hypothesis can be made stating that pesticide metabolites can also affect oxidative stress and important renal enzymes. In order to study this hypothesis, metabolites of acephate (AC), chlorpyrifos (CP), diazinon (DZ), dimethoate (DM), fenthion (FN), fenamiphos (FM), phenthoate (PH), profenofos (PF), quinalphos (QP), imidacloprid (IM), and Glyphosate (GP), Figure 1 shows the metabolites of each pesticide, which are considered in this paper. Appendix A indicates the names of the pesticides and metabolites considered in the present study. Since mostly marketed and used pesticides are racemic mixtures, chirality of pesticide structures were not considered.

Even though there is a lot of research conducted on CKDu, a molecular level explanation on CKDu has not been put forward. In the present study, a computational approach was used in order to study the interaction of a selected pesticide and their metabolites with the above-mentioned target proteins that are involved in human renal function. Further, this study investigates the effects of pesticide metabolites on CKDu, which is an uncovered area.

## 2. Materials and Methods

### 2.1. Ligand Preparation

The structures of pesticides and their metabolites were prepared using Avogadro software [42], and all the electronic structure calculations were performed using Gaussian 09 software [43]. The structures were optimized using density functional theory (DFT) method with B3LYP functional [44,45] and 6-311G++(d, p) basic set. The vibrational frequencies of each molecule were observed after optimization to identify any imaginary frequencies, and imaginary frequencies were not observed for the considered structures, indicating that the geometries are at energy minima. 

### 2.2. Protein Structure Preparation

AMPK, PKC, GLS, ASK1, AChE, CP450, and GST proteins were used in the present study. The starting coordinates for each protein were taken from X-ray crystallographic structures: PDB IDs 6C9G [46] for AMPK, 2I0E [47] for PKC, 3VP1 [48] for GLC, 3VW6 [49] for ASK1, 6NEA [50] for AChE, 3NXU [51] for CP450, and 1AQW [52] for GST. If there were missing amino acids, the following steps were followed to add the missing amino acids and build a new model. First, the RCSB protein data bank was searched for proteins with ≥80% similar sequence to the protein of interest using the BLAST server [53]. Next, the Modeler 9.22 software package [54] was used to add the missing AA and build a new model. The model with least Discrete Optimized Protein Energy (DOPE) score was selected for verification of the models. Protein structures were subjected to testing. Verify3D [55], ERRAT [56], and PROCHECK [57], under the SAVES v5.0 server, together with ProSA [58], were used to evaluate the validity of the protein structures. The results of these tests are tabulated in Appendix A. Finding the protein active sites and substrate-binding sites was done using the GASS-WEB server (Appendix A) [59]. All of the above-found amino acid sequences were subjected to NCBI-VAST database search [60] in order to confirm results.

### 2.3. Molecular Docking Details

Protein and ligand pdbqt files were loaded to AutoDock Tool 1.5.6 AutoDock 4.2 [61] (.dpf), and Autogrid 4.2 (.gpf) files were generated. Autogrid 4.2 was used to generate the grid parameter files and map files. The genetic algorithm parameters were set as follows; the number of genetic algorithms (GA) runs: 100, population size: 300, the maximum number of evaluations: 25,000,000, and the other setting were set to default values. AutoDock 4.2 was used for docking, and docking log files (.dlg) were generated. Appendix A shows AutoDock 4 grid parameters.

### 2.4. Molecular Dynamics Simulation Details

Protein-pesticide metabolite complexes with the best negative binding energies from docking studies were selected to perform molecular dynamics (MD) simulations in order to get further details about the pesticide metabolite binding with the proteins. Further, in order to perform a comparison, MD simulations were carried out for the protein–parent pesticide molecule complexes of the pesticide metabolites which were considered for MD simulations and also for the apo-enzymes. Table 1 shows the proteins, and corresponding pesticide and metabolite molecules considered for the MD simulations. In the case of PKC, M4 had the most negative binding energy. However, according to Casida et al. [62], Imidacloprid has higher potential to cause dysfunction of PKC. Therefore, PKC-6CIPHD and PKC-IM complex were used for the MD simulations.

MD simulations for all the protein-ligand complexes and apo-enzymes were performed using GROMACS (version 2016.3) [63] software package, using CHARMM36 force field [64] and TIP3P water model [65]. The protein-ligand complex was centered in a periodic box with a minimum distance of 1.0 nm between protein and any side of the box. The system was solvated with water, and Na^+^ and Cl-ions were added, replacing solvent molecules, in order to neutralize the systems at a 0.15 M salt concentration. The box sizes and number of water molecules for each system are mentioned in Appendix A.

The LINCS bond length constraint algorithm [66] was used to constraint bond lengths. Particle Mesh Ewald summation [67] was used for electrostatic interactions, and grid spacing of 0.12 nm combined with an interpolation order of 4 was used for long-range interactions. For van der Waals interactions, a cut-off of 1.4 nm was used. Energy minimization was performed using steepest descent algorithm [68]. The systems were gradually heated from 50 K to 300 K throughout a 200 ps time. Finally, the production runs were done in NPT ensembles at 300 K using V-rescale thermostat [69] and at 1 bar using Berendsen barostat [70]. Results of the all the simulations were obtained after 50 ns production runs with 2 fs time steps, and three multiple trajectories were generated for the each system using different randomly assigned initial velocities. 

### 2.5. Validation of the Theoretical Approach

Inhibitors and activators are commonly found in protein structures are deposited in the RCSB protein data bank. The structures of the inhibitors or the activators were taken from the PubChem database, and they were optimized using the same level of theory, which was used for the optimization of pesticides and their metabolites. These optimized structures were used as ligands for docking analysis. The activators and inhibitors were docked to their corresponding proteins using the same docking protocol which was used for the docking of protein and pesticides or their metabolites. The binding energies of the docked conformation were analyzed using AutoDock Tools 1.5.6. The binding residues of the docked conformations were compared to the binding residues stated in literature as a validation method for the docking protocol used in the present study. Further, as mentioned before, multiple trajectories were generated for each protein and pesticides or their metabolites systems during MD simulations, in order to confirm the possibility of obtaining consistent results and also for the statistical analysis.

### 2.6. Analysis

The docking results were analyzed using AutoDock Tool 1.5.6 to examine the binding energies. The binding pocket of the ligand was analyzed using Ligpot^+^ V.2.2 [71]. The protein-ligand interaction profiler [72], an online web-based service, was also used to validate the binding residues found by Ligpot^+^ V.2.2.

The resulting 50 ns MD trajectories of systems were analyzed by plotting radius of gyration (Rg) and root mean square deviation (RMSD) using GROMACS software package, to evaluate the stability of the protein-ligand complexes throughout the simulation time. In order to explain the better binding energies between protein and considered pesticide metabolites, with the help of atomistic details, hydrogen bond analysis, root mean square fluctuations (RMSF), and solvent accessible surface area (SASA) calculations were performed using GROMACS software package. In order to quantify the strength of the interaction between protein–ligand complexes, the non-bonded interaction energies were calculated between protein and ligands, using GROMACS software package. The binding free energy summation of the polar, non-polar energies, and non-bonded interaction energies (Vander Waals and electrostatic interaction) was calculated using the MM-PBSA method [73]. Further, interaction entropies were calculated for the protein-ligand complexes following the interaction entropy paradigm by Duan et al. [74].

In order to investigate the eigenvectors which play an important role in protein motions during ligand binding, principal component analysis (PCA) was performed using Bio3D package [75].

Rg, RMSD, RMSF, SASA, and hydrogen bond analysis plots were obtained for the individual trajectories, and they were averaged over multiple trajectories. Hydrogen bond lifetimes and radius of gyration values were averaged over multiple trajectories, and standard deviations were calculated and reported. Since the consistency was observed over multiple trajectories, PCA results were presented for a single trajectory in the present study. 

## 3. Results

### 3.1. PDB Structure Refinement and Model Validation

PDB structure refinement was done for the AMPK, PKC, ASK1, AChE, and CP450 proteins by adding missing amino acids. These amino acids have to be added as these flexible regions may be involved in the active site of enzymes. Modeler model building software is used to build new models. Modeler needs template structures for its function. These templates have to have ≥ 80% similarity to the protein of interest. The BLAST server was used to find the homology sequences, and the results of the BLAST search are mentioned in Appendix A. The model with the least DOPE score was selected and used for the model validation. The three-dimensional models of the studied eight proteins were validated by VERIFY3D score [55]. VERIFY3D analyzed the compatibility of an atomic model (3-D) with its own amino acid sequence. Seven out of eight models passed the VERIFY3D test. The passed residues have a score of greater than 0.2 and this dictated the quality of the model (Appendix A). The ERRAT server [56] is used for analyzing the statistics of non-bonded interactions between different atom types, and scores greater than 50 are normally acceptable. For all eight models, ERRAT score varies from 85.62 to 99.00. (Appendix A), which fall within the normal range for high quality model. ProSA [58] is widely used to check for potential errors in 3-D protein structure models. The Z-score shows the cumulative reliability of the model and analyses the variance of the cumulative structural energy from the distribution of energy from spontaneous conformations. The ProSA score was negative for the modeled protein, indicating its validity. ProSA score varies from −3.8 to −9.93 (Appendix A). Further, the geometries of 3-D structures were evaluated using Ramachandran plot calculations with PROCHECK [57]. Stereochemical evaluation of backbone Psi (Ψ) and Phi (Φ) dihedral angles of five human proteins were revealed in different percentages, i.e., 87.7–94.4%, 5.1–11.2%, and 0.5–1.1%. Residues were diminished within the most favored regions, additionally allowed regions, and generously allowed regions, respectively. The dihedral angles revealed that some residues like 0.0–0.3% are on disallowed regions of Ramachandran plot. The models have a normal distribution of residue types over the inside and the outside of the protein structures. Therefore, it can be stated that the overall results from VERIFY3D score, ERRAT score, ProSA score, and PROCHECK validated the selected protein models. 

### 3.2. Protein Active Site Prediction Analysis

Finding the protein active sites and substrate-binding sites was done using the GASS-WEB server [59]. The GASS-WEB server consists of active-site models and their respective Protein Data Bank structures. GASS-WEB uses the Catalytic Site Atlas (CSA) models of 1691 catalytic site. The database is also composed of 1819 binding site templates from CSA and 23,318 enzymes from the NCBI-VAST non-redundant database. This indicates the reliability of the result when it is proven to be correct by the NCBI-VAST database search. Appendix A shows the verified results of active site analysis. Three possible catalytic and binding site options with the least fitness score were recorded to increase the possibility of getting the true binding site. 

Tuck and coworkers conducted an analysis of the active site of CP450 [76]. They found the residues PHE87, TYR96, ILE395, VAL295, VAL396, THR252, VAL247, LEU244, and THR185 to be in the active site of CP450. Out of these 9 amino acids, 5 of them were predicted by GASS-WEB server. The active site of GLS has been, analyzed by Thangavelu. K and coworkers [77]. SER286, TYR249, ASN335, GLU381, ASN388, TYR414, TYR466, and VAL484 are the residues found to be in the active site [77]. These residues werealso predicted correctly according to GASS-WEB server results. The active site residues of GST were reported as SER65A, ASP98B, GLN64A, LEU52A, TRP38A, LYS44A, and GLN51A [52]. Residues TRP279, TYR334, ASP72, TRP84, SER200, PHE331, and TYR121 are found to be in the active site of AChE [78]. The correct prediction of the binding residues dictates the accuracy of the GASS-WEB server. Therefore, it can be used to predict the active site of enzymes whose active site data are not found in literature. The compatibility of theoretical predictions with the real world further dictates the high quality of protein models. ATP binding sites of ASK1 are GLN756, LEU686, GLY687, MET754, PHE823, VAL810, and ASN808 [79]. The Autophosphorylation sites of ASK1 are THR813, THR838, and THR842 [79]. These three sites are important in the regulation of ASK1 [79]. GLU725, LYS709, SER826, ASP822, PHE823, and GLU837 are parts of the activation site of ASK1 [79]. GO 6983 is an inhibitor that binds to the active site of PKC. GLU421, THR404, VAL356, LEU348, ASP484, ALA483, and PHE353A are residues involved in the binding of G0 6983 [80]. The other important residues of the catalytic domain of PKC are THR500, THR641, and SER600, which are autophosphorylation sites of PKC and PHE629, which is an ATP binding residue [80]. Though diacylglycerol and phosphatidylserine binds residues are important, they are found in the C2 domain of PKC and not the catalytic domain [81]. AMPK has may important regions. It has three AMP binding regions, autophosphorylation sites at THR12 and 148, regulatory spin at LEU68, LEU79, HIS137, and PHE158 [82]. The active site residues of AMPK are LYS47, LEU29, VAL26, ALA45, GLU96, TYR97, VAL98, SER99, LEU148, ALA158, ASP159, GLU145, ASN146, and GLU145 [83]. PKD 2 has an ATP binding site, which controls the phosphorylation process [84]. Pyrazolo[3,4-*d*]pyrimidine derivative binds to the ATP binding site of PKD 2. The active site location is very important when analyzing docking results. When the docked location of a particular ligand is close or at the active site, the effect of that particular ligand on the protein function is more pronounced.

### 3.3. Molecular Docking

When considering all the proteins, the average binding energies of the eleven pesticides and metabolites range from −4.34 kcal/mol to −6.69 kcal/mol. The highest average binding energy was shown by FM and its metabolites. The lowest average binding energy was shown by AC and its metabolites. This variation of binding energies can be clearly seen in Figure 2. Appendix A shows the residues responsible for ligand-protein interactions and the types of dominating interactions by each residue. The most stable interaction was observed between FMS metabolite and AChE (−8.22 kcal/mol). The second most stable interaction was observed between M4 metabolite and AChE (−8.17 kcal/mol), and the third most stable docking was observed between IM pesticide molecule and CP450 with the binding energy of −7.88 kcal/mol. The fourth strongest interaction was observed between FMS and PKC with the binding energy of −7.83 kcal/mol, and the fifth strongest was observed between GLS and FMS metabolite (−7.75 kcal/mol). It is interesting to see that, out of the above discussed five cases, the metabolites were observed in four of them. The docked positions for the above mentioned five cases with the highest binding energies can be seen in Figure 2.

MED, DEP, DETP, MMP, DMP, and DMTP, which are of monoalkyl and dialkyl phosphates or monoalkyl and dialkyl thiophosphate metabolites of AC, CP, DZ, DM, PH, and QP show the average binding energies of −3.80 kcal/mol, −4.66 kcal/mol, −4.18 kcal/mol, −4.31 kcal/mol, 4.37 kcal/mol, and −4.05 kcal/mol, respectively. This variation of binding energies can clearly be seen in Figure 3. ARG, ASN, GLN, GLU, TRP, and LYS are major interacting amino acids found in the binding pocket. The major type of interactions with the binding pocket are salt bridges and H-bonding. It can be seen in Figure 3, when the binding energies of pesticide GP pesticide and its metabolite AMPA are compared, there is an average of 0.84 kcal/mol difference between binding energies of AMPA and GP. GP forms H-bonding, and salt bridges, as the dominant interactions with amino acid residues of the binding pocket, whereas AMPA form H-bonding as the dominant interaction. A detailed description of the binding pockets can be seen in Appendix A.

According to Figure 3, when the binding energy of AC pesticide (average binding energy = −4.05 kcal/mol) compared with its metabolite MED (average binding energy = −3.86 kcal/mol), AC has a higher binding affinity with all the proteins considered. The number of residues involved in AC binding is higher when compared with MED. It can be seen in Figure 3, when binding energies of CP pesticide compared with its metabolites (TCP, DEP, and DETP), CP has the highest binding affinity, except in one case. TCP manages to outbound with respect to enzyme PKC. As discussed above, DEP and DETP, which are dialkyl phosphates and thiophosphates, have the least bind affinities. CP has an average binding energy of −5.91 kcal/mol, whereas TCP has an average binding energy of −5.2 kcal/mol. CP forms H-bonding, halogen bonding, and п-п stacking as the major types of interactions. TCP metabolite forms H-bonding and halogen bonding. Further, hydrophobic interactions can be seen between aromatic amino acids and TCP aromatic rings.

As can be seen in Figure 3, binding energy variation trend of DZ pesticide and its metabolites is somewhat similar to that of CP. In this case, DETP and DEP metabolites have the least binding energies. On average, DZ pesticide molecule has the highest binding energy, which is equal to −6.02 kcal/mol, whereas IMP metabolite has an average binding energy of −5.68 kcal/mol. When considering enzyme PKC, IMP metabolite has −0.88 kcal/mol higher binding energy than DZ pesticide molecule. DZ forms H-bond, π-π stacking, π-cation interactions, and hydrophobic interactions. On the other hand, IMP metabolite forms H-bonding, π-π staking, salt bridges, and hydrophobic interaction as the major type of interactions. According to Figure 3, when analyzing the variation of binding energies of DM and its metabolites MMP and DMP, an interesting change of the trend can be seen. In all of the above-discussed cases, di and mono alkyl phosphates have low binding energies compared to parent pesticide. But, in this case, DMP and MMP also have high binding energies compared to DM pesticide. The average binding energies of DM, MMP, and DMP are −4.47 kcal/mol, −4.37 kcal/mol, and −4.31 kcal/mol.

FN pesticide forms 4 different metabolites. According to Figure 3, all the 4 metabolites of FN (FX, FXS, FNS, and FXSX) have higher binding affinity than the parent pesticide. The average binding energies of FN, FX, FNS, FXS, and FXSX are −5.78 kcal/mol, −6.02 kcal/mol, −6.80 kcal/mol, −6.86 kcal/mol, and −6.47 kcal/mol, respectively. The order of increasing binding affinity of proteins to FN and its metabolites are PDK 2 < AMPK < CP450 < GST < ASK1 < PKC < GLS < AChE. When FN change to FXS, the binding energy increased by −1.1 kcal/mol. When comparing FNS and FNSX, the binding energy decreased by 0.39 kcal/mol. The same pattern can be observed with respect to FX and FXS. When comparing FN with FXSX, an increment of −0.69 kcal/mol can be seen. FN and metabolites make H bonds, salt bridges, and π-π staking. Due to the presence of aromatic ring, all the metabolites make hydrophobic interactions with hydrophobic residues of the binding pocket. FM also has 4 metabolites (FMS, FMSX, DFS, and DFSX), and it can be seen in Figure 3 that all the metabolites outbound the parent pesticide molecule. The average binding energies of FM, FMS, FMSX, DFS, and DFSX are −6.34 kcal/mol, −7.15 kcal/mol, −6.74 kcal/mol, −6.66 kcal/mol, and −6.50 kcal/mol, respectively. The order of increasing binding affinity of proteins to FN and its metabolites are GLS < PKC < ASK1 < GST < CP450 < AMPK < AChE. When comparing FM with FMS, the binding energy increased by 0.81 kcal/mol. When we compare FMSX with FMS, the binding energy decreased by −0.40 kcal/mol. When FMS compared with DFS the binding energy decreased by 0.6 kcal/mol. The same trend can be seen when FMSX compared to DFSX. FM makes H-bonds and π-cation interactions. FMS and FMSX make H-bonds and π–π stacking. DFS and DMSX have H-bonding and DFSX has salt bridges and π-cation interaction. As FM and all the metabolites have an aromatic ring, it has formed hydrophobic interactions with hydrophobic patches of the binding pocket.

When analyzing PH pesticide and one of its metabolites, PC, neglecting DMTP, which is another metabolite, there is very little variation of binding energies between them (Figure 3). The average binding energies of PH, PC, DEMPA, DEMP, DEMPOA, DEMPO, and DMTP are −5.88 kcal/mol, −6.13 kcal/mol, −6.04 kcal/mol, −6.05 kcal/mol, −6.27 kcal/mol, −6.00 kcal/mol, and −4.05 kcal/mol, respectively. The pesticide molecule PC has −0.25 kcal/mol higher binding energy than its metabolite PH. When comparing PH with its metabolite DEMP, this metabolite has higher binding energy than PH. When DEMPOA is compared with DEMPA, binding energy increased by 0.23 kcal/mol. On the other hand, when DEMPO is compared to DEMP the binding energy decreased. PH and all its metabolites expect DMTP makes H-bonding and salt bridges with the binding pocket. Amino acid ARG makes π-cation interaction with the aromatic ring of PC. π-π staking can also be seen to be formed with the aromatic ring of the ligand and amino acids, such as TRP, HIS, and PHE. As can be seen in Figure 3, the average binding energy of PF, M1, M3, and M4 are −6.26 kcal/mol, −6.44 kcal/mol, −6.01 kcal/mol, and −6.54 kcal/mol. Therefore, it can be stated that M1 metabolite has higher average binding energy compared to the PF parent pesticide. M1 is the only molecule out of PF category to make salt bridges. H-bonding and π-π staking residues of M1 and M3 binding are similar to that of PF. In M4, in addition to H-bonding and halogen bonding, Л-cations interactions can also be observed with HIS. The nonpolar regions of the four molecules can make hydrophobic interactions.

The only metabolite of QP pesticide forms other than DEPT is HQ. According to Figure 3, the average binding energies of QP and HQ are −6.44 kcal/mol and −5.69 kcal/mol. Metabolites of IM (IG, 6-CIPHD, and 6-CINA) have lower binding affinities compared to the parent pesticide molecule. Average binding energies of IM, IG, 6-CIPHD, and 6-CINA are −6.67 kcal/mol, −6.31 kcal/mol, −6.39 kcal/mol, and −5.96 kcal/mol, respectively. IM makes H-bond, salt bridges, and halogen bonding. In addition to the above-mentioned interactions, π-π staking and π-cation interactions can be seen in IG. Binding residues involved in 6-CIPHD and 6-CINA are similar to that of IM and IG.

### 3.4. Validation of the Docking Approach

Inhibitors and activators are commonly found in protein structures are deposited in the RCSB protein data bank. The activators and inhibitors were docked to their corresponding proteins using the same docking protocol which was used for the docking of protein and pesticides or their metabolites. The theoretical model is validated by comparing the theoretical binding residues with the binding residues from literature. Table 2 contains the comparison data. As shown in the data, there is very little deviation between the two locations. This confirms the validity of the theoretical model.

### 3.5. Analysis of the Docked Location

To have a clear idea about the effect of particular ligand on a protein, the docked location of the ligand has to compare with the important sites of the protein. These important sites can be the active site, reactivator site, and inhibition site. When the ligand binds to the active site of the enzyme it can acts as competitive inhibitors. Figure 4A shows the active site and reactivator site of AChE protein, and it can be seen that, in AChE, the active site and the reactivator sites are situated somewhat closer to one another. In fact, TYR341 and PHE338 are both in the active site and reactivator site. As can be seen in Appendix A, AC, MED, CP, DEP, DETP, DZ, DM, FN, FX, FNS, FXS, FXSX, DFD, DFEX, FM, FMS, FMSX, PH, DEMP, DEMPO, PF, M3, M4, 6-CIPHD, QP, and HQ were bound to the active site of the enzyme. FM, FN, and their metabolites were bound right inside the binding pocket more perfectly than other ligands. The binding patterns indicates that TYR341 and PHE338 are very important for binding. TCP, IMP, MMP, DMP, DEMPA, PC, DEMPOA, M1, IG, IM, 6-CINA, GP, and AMPA are bound to other sites away from the active site and reactivation side. Even though AMPA was bound away from the active site, the bound location was closer to the active side than in the case of GP. According to Appendix A, another interesting observation is that 6-CIPH, which is a metabolite of organochlorine, can also bind to the active site of AChE as organophosphates with roughly similar binding affinity (Figure 3).

When analyzing the results of CP450 protein interaction with pesticide, the binding pocket of Ritanovir and active sites can be taken as reference points. As can be seen in Figure 4B, these two sites are situated somewhat close to each other. The two pockets do not have any common residues. None of the ligands bound to the active site. But CP binds to a site, which has high proximity to the active site. TCP, DZ, PF, PC, M4, and QP manages to bind in the vicinity of the inhibitory site. But none of them were able to fully occupy the inhibitory pocket. Ligands other than the ones mentioned above binds to CP450 at very distal locations to the active site and inhibitory site. When considering the binding location of ligands to GLS protein, for pesticides and metabolites, binding site was different than the active site (Figure 4C). However, FN, FM, PH, CP and their metabolites and QP bind to a site proximal to the active site. On the other hand, DM, MMP, DMP, GP, AMPA, 6 CINA, 6-CIPHD, HQ, DZ, and DMP bound to distal locations from the active and inhibitory sites. Another interesting observation is that AC manages to bind to the inhibitory site perfectly, and IG manages to bind near the inhibitory site.

GST protein is inhibited by Ethacrynic Acid (EA) [85]. The binding residues of GST which participated in the binding of EA are GLN39, ILE103, and PHE8. Figure 4E shows the inhibitory site and active site of GST protein, along with the pesticide and ligand binding. Ligands TCP, DETP, DFSX, PH, IG, 6-CIPHD, QP, and HQ bind to the active site or a site which is very proximal to the active site. MED, DM, FNS, FN, DEMP, DEMPO, DEMPOA, and AMPA bound to sites which are distal when compared with the above instance. However, the binding site may be close enough to cause problems in the active site. IM and AC binds to the inhibition site. DEP, CP, MMP, DMP, FX, FXS, FXSX, FM, FM, FMS, FMSX, DFS, DMTP, PC, PF, M1, M3, M4, 6-CINA, and QP binds to sites which are far away from the active site and inhibition site. ASK1 has 4 important sites: the active site, ATP binding site, autophosphorylation site, and inhibition site [79]. These sites are shown in Figure 4F. The active site, ATP binding site and autophosphorylation site are located in the same pocket. A ligand bound to this pocket can effectively block the active site and ATP binding site or may act as an inhibitor [79]. FN and FM pesticides and their all the metabolites bind perfectly to the above mentioned pocket. Further PH pesticide and its metabolite DEMP, PF pesticide and its metabolite M3, and few other metabolites, namely TCP, IMP, and HQ bind perfectly to the same pocket. DM, QP, PC, DEMPA, DEMPO, and DEMPOA bind proximal to the opening of the above mentioned pocket. AC, MED, DZ, MMP, DMP, DMTP, CP, DEP, DETP, and all the metabolites and parent molecule of IM and GP bind to sites which are far away from the active site, and their effect is minimal.

When analyzing docked locations of PKC protein, it can be observed that, out of all the ligands, only FN, FM, and 6-CIPHD are bound to the active site of PKC (Figure 4D). AMPA bind to a location that is proximal to the active site. All the ligands except the ones which are mentioned above bound to locations other than the active site, the three autophosphorylation sites, and the ATP binding site. As mentioned earlier, there are many functionally important sites in AMPK. The active site, binding site, activator site of AMPK protein, and pesticide bindings are shown in Figure 4G. Out of all the ligands, only TCP and QP bind to the active site of AMPK. AC, DETP, CP, DZ, TMP, DM, IM, 6-CINA, and all the metabolites and parent pesticide molecules of FN, PH, PF, and FM, except DFSX, bind to sites which are in very close vicinity to the AMP binding sites. MED, DEP, MMP, DMP, DFSX, IG, 6-CIPHD, HQ, GP, and AMPA did not bind in the vicinity of any important sites.

### 3.6. Molecular Dynamics Simulations

Protein-pesticide metabolite complexes with the best negative binding energies were subjected to molecular dynamics (MD) simulations in order to get further details about the pesticide metabolite binding with the proteins. Further, in order to perform a comparison, MD simulations were carried out for the protein-parent pesticide molecule complexes of the pesticide metabolites, which were considered for MD simulations. Simulation details are explained in the Methodology section, and the three multiple trajectories were generated for protein-metabolite systems using different randomly assigned initial velocities, in order to confirm the possibility of obtaining consistent results and also for the statistical analysis. All the graphs and the values available below are results which have been averaged over multiple trajectories. All the studied protein-ligand systems showed good stability throughout the 50 ns simulation time. All the calculations were performed using 50 ns simulation trajectories.

#### 3.6.1. Root Mean Square Deviation (RMSD)

The stability of the protein-ligand complex systems with highest binding energies according to docking, was analyzed based on the root mean square deviation (RMSD). Figure 5 shows the RMSD plots for protein-pesticide metabolite complexes with the best negative binding energies (red), along with the relevant protein-parent pesticide complex (black) and apo-enzyme (blue). RMSD plots for protein-metabolite complexes are the average of three trajectories with different initial velocities. According to RMSD, all the protein-ligand complexes show stability in 50 ns simulation. In the case of AChE-FMS protein-metabolite complex, average RMSD is 0.21 (±0.02) nm, whereas the relevant protein-parent pesticide complex (AChE-FM complex) shows average RMSD of 0.21 (±0.02) nm, which are similar to RMSD for apo-enzyme 1.91 (±0.02) nm. AMPK-FNS protein-metabolite complex has average RMSD 0.47 (±0.08) nm, and the relevant protein-parent pesticide complex (AMPK-FN complex) has a better average RMSD of 0.45 (±0.08) nm. Even though RMSD for protein-ligand complexes are fairly high, they are in the same range as RMSD of apo-enzyme AMPK (0.41 ± 0.06 nm). In the case of GST–6-CIPHD protein-metabolite complex, average RMSD is 0.20 (±0.02) nm, whereas the relevant protein-parent pesticide complex (GST-IM complex) shows average RMSD of 0.22 (±0.03) nm, and apo-enzyme GST shows similar RMSD of 0.18 (±0.02 nm). For ASK1-FMS protein metabolite complex, average RMSD is 0.24 (±0.02) nm, which is similar to the ASK1-FM relevant protein–parent pesticide complex (0.21 ± 0.02 nm) and apo-enzyme ASK1 (0.29 ± 0.02 nm). PKC–6-CIPHD protein-metabolite complex has a better average RMSD of 0.24 (±0.02) nm when compared with the relevant protein-parent pesticide complex (PKC-IM complex) average RMSD of 0.29 (±0.07) nm and apo-enzyme PKC RMSD of 0.28 (±0.02 nm).

The RMSD results of protein-ligand complexes considered in MD simulations reveal the stability of both the protein-pesticide complexes and protein-metabolite complexes. However, GST-IM, GST-6-CIPHD, and PKC-IM complexes shows significant drifts in RMSD. Therefore, per-frame backbone RMSD matrix calculation was performed for these three complexes and AMPK-FN/FNS complexes which show high RMSD. Backbone RMSD matrix calculation results can be seen in Appendix A. RMSD matrix with the smallest pairwise RMSD are in blue, while the largest RMSD is in dark red. It is visible according to per-frame RMSD matrix that, in the case of GST-IM, GST-6-CIPHD, and PKC-IM complexes, where the significant RMSD shifts can be seen, structures shift to a state which is a drifted structure relative to the reference structure. Therefore, an argument can be made that these states can be alternative stable states.

#### 3.6.2. Radius of Gyration (Rg)

Since radius of gyration is an indicator of protein compactness, which can assess the protein-ligand complex stability/ instability [86], the plots of Rg as a function of time for all the protein-metabolite complexes (red), protein-pesticide complexes (black), and apo-enzymes (blue) which were studied by MD simulation are shown in Appendix A. The average Rg values of protein-ligand complexes and native protein are shown in Table 3, and a similarity of Rg values was observed for both the protein-pesticide complexes and protein-metabolite complexes compared to the native protein. This indicates the stability of both the protein-pesticide complexes and protein-metabolite complexes.

#### 3.6.3. Root Mean Square Fluctuation (RMSF)

Protein dynamics by residue were examined by calculating RMSF. The RMSF per residue values are presented in Figure 6 for the proteins in protein-metabolite complexes (red) and in protein-pesticide complexes (black). Figure 6A shows RMSF for AChE-FMS protein-metabolite complex and its protein-parent pesticide complex AChE-FM. It can be seen that residues GLU3, ASP4, ARG164, PRO258–ASN164, PRO289–GLU291, ASP383–GLU388, and ARG492–LYS495 of the protein AChE are highly flexible in the presence of both the FM pesticide molecule and FMS metabolite and also in apo-enzyme. Figure 6B shows RMSF for AMPK-FNS protein-metabolite complex, its protein-parent pesticide complex AMPK-FN, and apo-enzyme AMPK, respectively. It can be seen that residues ALA369A–LYS393A, GLN79B–GLY81B, TYR126B–ARG135B, SER24C, and ASN25C of the protein AMPK show large fluctuations with both the FN pesticide molecule and FNS metabolite and also in apo-enzyme. Figure 6C shows RMSF for ASK1-FMS protein-metabolite complex and its protein-parent pesticide complex ASK1-FM, respectively. It can be observed that ASK1 protein shows similar dynamics in the presence of both the FM pesticide molecule and FMS metabolite, where residues ASP8–ASN10, GLU43–TYR48, ILE162–PRO164, GLY189, GLU213, LEU214, and LYS269 of the protein ASK1 are highly flexible. According to the Figure 6C, which shows RMSF for GST–6-CIPHD protein-metabolite complex and GST-IM protein-parent pesticide complex, respectively, protein residues GLU36–LYS44, LYS81–GLN83, TYR108–ASN117, GLN135–GLY140, and ASN206-GLN209 of GST protein are highly dynamical, with both the pesticide molecule and metabolite and also in apo-enzyme. Figure 6E shows RMSF for PKC–6-CIPHD protein-metabolite complex and its protein-parent pesticide complex PKC-IM. It can be seen that residues LEU19–ASN23, GLY31–PHE33, ARG41–GLU46, MET93, GLU224–GLU228, GLY270, and LYS300–PRO317 of the protein PKC are highly flexible in the presence of both the IM pesticide molecule and 6-CIPHD metabolite and also in apo-enzyme.

However, none of the above-mentioned protein residues for any of the proteins are involved in ligand interactions according to Appendix A, which shows binding pocket protein residues within 7 nm cut-off from ligands. Further, from RMSF results, it is observable that all the proteins considered in MD simulations show similar flexibilities to the apo-enzyme in the presence of metabolites and their relevant pesticide molecules.

#### 3.6.4. Solvent Accessible Surface Area (SASA)

Since SASA parameter measures the proportion of the protein surface which can be accessed by the water solvent, it can be used to predict the extent of the conformational changes that occurred during protein-ligand interactions. Figure 7 shows the plots of SASA values versus time for proteins in protein-metabolite complexes (red) and in protein-pesticide complexes (black). In the case of AChE-FMS protein-metabolite complex, average SASA is 40.79 (±0.82) nm^2^, whereas the relevant protein-parent pesticide complex (AChE-FM complex) shows a similar average SASA value of 39.07 (±0.84) nm^2^; AMPK-FNS protein-metabolite complex has average SASA value of 40.73 (±0.82) nm^2^, and the relevant protein-parent pesticide complex (AMPK-FN complex) has a similar average SASA value of 39.07 (±0.84) nm^2^. For ASK1-FMS protein metabolite complex, average SASA is 32.92 (±0.52) nm^2^, which is similar to the ASK1-FM relevant protein–parent pesticide complex (32.18 ±0.59 nm^2^). In the case of GST–6-CIPHD protein-metabolite complex, average SASA value is 39.81 (±0.47) nm^2^, whereas the relevant protein-parent pesticide complex (GST-IM complex) shows a similar average SASA value of 38.81 (±0.67) nm^2^. PKC– 6-CIPHD protein-metabolite complex has a similar average SASA value of 28.95 (±0.42) nm^2^ when compared with the relevant protein-parent pesticide complex (PKC-IM complex) average SASA value of 28.47 (±0.66) nm^2^. Therefore, it can be stated that all the protein-metabolite complexes considered in MD simulations show similar dynamics as relevant protein-parent pesticide complexes.

#### 3.6.5. Hydrogen Bond Analysis

Hydrogen bonds play a significant role in the protein ligand binding. Figure 8 displays the number of hydrogen bonds between studied protein and ligand during 50 ns simulation time. It is clearly visible that metabolites (red) have higher number of hydrogen bonds with the protein (specifically AChE-FMS, AMPK-FN, ASK1-FMS, and PKC–6-CIPHD protein-metabolite complexes), compared to corresponding protein-parent pesticide complexes (black). In the case of GST–6-CIPHD protein-metabolite complex, it has similar number of hydrogen bonds as GST-IM protein-parent pesticide complex.

These number of hydrogen bonds results analysis is helpful in explaining the higher binding energies of protein-metabolite complexes compared to protein-parent pesticide complexes: AChE-FMS (−8.22 kcal/mol) compared to AChE-FM (−7.32 kcal/mol), AMPK-FNS (−6.82 kcal/mol) compared to AMPK-FN (−5.34), and ASK1-FMS (−7.24 kcal/mol) compared to ASK1-FM (−6.23 kcal/mol). In the case of PKC– 6-CIPHD protein-metabolite complex, where it shows similar binding energy (−7.18 kcal/mol) to its protein-parent pesticide complex, PKC-IM (−7.28 kcal/mol), both the metabolite complex and pesticide complex, show similar number of hydrogen bonds up to around 25 ns of simulation time, and, after that, IM pesticide molecule does not show hydrogen bonds with PKC protein, where PKC-6-CIPHD protein-metabolite complex manages to show hydrogen bonds throughout the 50 ns simulation time. This breaking of hydrogen bonds between PKC protein and IM pesticide molecules explains the RMSD variation of PKC-IM complex (Figure 8D). In the case of GST–6-CIPHD protein-metabolite complex, even though it shows higher binding energy (−7.54 kcal/mol) compared to GST-IM protein-parent pesticide complex (−6.22 kcal/mol), number of hydrogen bonds are similar for both complexes throughout the simulation time. Overall, comparison of these observed hydrogen bonding parameters, along with the binding energies, indicate that above-mentioned metabolites can be bound to the corresponding proteins, which are linked to the CKDu more effectively and tightly, compared to relevant parent pesticide molecules.

#### 3.6.6. Principal Component Analysis (PCA)

The PCA analysis (Figure 9) was carried out to investigate the collective motion of the protein-ligand complexes. For AChE protein, the first 20 vectors account for 71.4% and 68.6% with FMS metabolite and FM pesticide, respectively, and, for AChE apo-enzyme, the first 20 vectors capture 67.2% collective motion. This indicates that the ligand binding, especially the FMS metabolite binding, increased the AChE protein collective motions. In the case of AMPK protein, the first 20 vectors represent similar percentages with FNS metabolite (91.2%) and FN pesticide (93.2%) to the apo-enzyme AMPK (92.5%). For ASK1 protein, the first 20 vectors are responsible for 82.5% and 72.0% of collective motions with FMS metabolite and FM pesticide, respectively, and, for ASK1 apo-enzyme, the first 20 vectors capture 84.4% collective motion, indicating FM pesticide binding has restricted ASK1 collective motions. In the case of GST protein, it can be seen that the first 20 vectors represent 76.3% with 6-CIPHD metabolite and 72.4% with IM pesticide. For the GST apo-enzyme, the first 20 eigenvectors capture 67.9% collective motions. This indicates that the ligand binding, especially the 6-CIPHD metabolite binding increased the GST protein collective motions. When we consider PKC protein, the first 20 vectors accounts 68.8% and 79.5% with 6-CIPHD metabolite and IM pesticide, respectively, and, for PKC apo-enzyme, the first 20 vectors capture 80.4% collective motion. This indicates 6-CIPHD metabolite binding restricted PKC collective motions.

Further, 2-D projection plot of principal component 1 (PC1) versus principal component 2 (PC2) are shown in Figure 10. It should be noted that AMPK and ASK1 protein complexes with metabolites occupies less phase than with parent pesticide molecules, revealing the presence of comparatively stable clusters in the case of AMPK-FNS and ASK1-FMS protein-metabolite complexes. Other protein-metabolite complexes considered in MD simulations do not show higher occupancies in phase, compared to protein-parent pesticide complexes and apo-enzymes. In the case of AChE-FMS complex along the PC2 plane, a clear change in conformational motions with two distinct clusters can be identified compared to the apo-enzyme AChE. A similar situation can be observed for the GST-6-CIPHD complex along the PC1 plane, compared to the apo-enzyme GST. These two situations explain the enhancement of collective motions among the first 20 eigenvectors for AChE-FMS and GST-6-CIPHD protein-metabolite complexes, compared to the corresponding apo-enzymes. In the case of PKC enzyme, clear changes in conformations can be seen for apo-enzyme and PKC-IM protein-pesticide complex. However, in PKC-6-CIPHD protein-metabolite complex, distinct clusters are not observable, explaining the observed collective motion restrictions among the first 20 eigenvectors.

Further, Appendix A indicates the variations in contributions from first three principal components (PC1, PC2, and PC3) to the RMSF of above-mentioned protein-metabolite complexes and protein-pesticide complexes. However, as mentioned earlier highly flexible protein residues (according to RMSF) of the protein-ligand complexes considered in MD simulations are not involved in the binding pocket (Appendix A).

#### 3.6.7. Interaction Energy

In order to validate the binding energies generated using molecular docking studies, a detailed analysis was performed regarding the calculation of the free energies of interaction responsible for the protein-ligand binding using Parrinello-Rahman parameter of GROMACS. Figure 11 shows the average short-range Lennard-Jones energy plot of protein-ligand complexes. It can be seen that for AChE-FMS protein-metabolite complex with comparatively higher docking binding energies has better interaction energies (−199.43 ± 10.30 kJ/mol) compared to the AChE-FM protein-parent pesticide complex (−180.81 ± 12.58 kJ/mol). AMPK-FNS protein-metabolite complex, which also shows comparatively higher binding energies, has better interaction energies (−180.26 ± 15.29 kJ/mol) compared to the AMPK-FN protein-parent pesticide complex (−109.18 ± 12.36 kJ/mol). A similar situation was observed between PKC-6-CIPHD protein-metabolite complex (−176.26 ± 21.63 kJ/mol) and PKC-IM protein-parent pesticide complex (−120.33 ± 17.33 kJ/mol). The ASK1-FMS complex showed lower interaction energy of than ASK1-FM complex; however, their interaction energies were virtually in the same range. A similar trend was shown by GST-6-CIPHD protein-metabolite complex, and it manages to show the interaction energy in the same range as GST-IM protein-pesticide complex. The interaction energy values validated the molecular docking results, showing that these pesticide metabolites bind to the corresponding proteins which are linked to the CKDu.

Further, the binding free energy summation of the polar, non-polar energies, and non-bonded interaction energies (Vander Waals and electrostatic interaction) was calculated using the MM-PBSA method, and interaction entropies were calculated for the protein-ligand complexes following the interaction entropy paradigm by Duan et al. [74]. Binding free energy and interaction entropy results are shown in Table 4. It can be seen from Table 4 that binding free energies for protein-metabolite complexes in the cases of AChE, ASK1, and AMPK are in a similar range as protein-pesticide complexes. Further, eASK1-FMS complex shows lower interaction energy than ASK1-FM protein-pesticide complex and ASK1-FMS protein-metabolite complex shows less binding energy than protein-parent pesticide complex. In the case of GST, GST-6-CIPHD protein metabolite complex has significantly more favorable binding free energy compared to GST-IM protein pesticide complex. The binding energy values, together with the interaction energies, indicates that pesticide metabolites bind to the corresponding proteins which are linked to the CKDu, revealing the toxicity of these pesticide metabolites.

## 4. Discussion

It has previously been proven that all pesticides have toxic effects on mammals, birds, fish, reptiles, and insects. To the best of our knowledge, so far, there are no much studies covering the interaction of pesticide and their metabolites with target proteins that are involved in human renal function. In the present study, a computational approach was used in order to study the effects of pesticide metabolites on CKDu, and, further, interactions of selected pesticides and their metabolites were studied using molecular docking studies and molecular dynamics simulations.

By analyzing and comparing the binding energies, an idea about the binding affinities of ligand relative to each other can be obtained. According to molecular docking studies, it is interesting to see that, out of five cases with the highest binding energies, the metabolites were observed in four cases: M4 with AChE protein and FMS with AChE, PKC, and GLS proteins. This gives the idea that metabolites can be more toxic than their parent compounds. Further, binding energy comparison between pesticides and their metabolites indicated the possibility of high toxicity of some metabolites compared to the parent pesticides. When the binding energies of pesticide GP and its metabolite AMPA are compared, a clear difference can be observed, indicating that AMPA as a more potent candidate for causing protein dysfunction when compared with GP, just as indicated in the literature [87]. However, it is hard to come to a direct conclusion as the binding pocket of GP and AMPA are different.

All four metabolites of FN (FX, FXS, FNS, and FNSX) have higher binding affinity than the parent pesticide. When considering the structure of FN and FX, ester O has changed to P=O and the mercaptan group has replaced with ester bond. This structural change as caused in increment of binding energy. This suggests that the P=O may be important for binding. When FN changes to FXS that S connected to the aromatic ring has oxidized to have two S=O groups, and the binding affinity is higher. When comparing FNS and FNSX, the S connected to an aromatic ring is reduced to have one OH, one H, and one Methyl group. In this case, the binding affinity decreased for FNSX. The same pattern can be observed with respect to FX and FXS. All this evidence suggests that P=O and S=O groups are key ingredients of binding. When comparing FN with FXSX, an increment of binding affinity can be seen due to oxidation of the above-mentioned S group.

Further, all four metabolites of FM pesticide (FMS, FMSX, DFS, and DFSX) outbound the parent pesticide molecule. When comparing FM with FMS, the binding affinity is higher with FMS due to oxidation of S. When S of FMS reduces to form FMSX, the binding affinity decreased. Even though FMSX has S-OH group which can make H-bond, the binding affinity decreased. This indicates that S=O group may be more important for binding than S-OH group. When FMS compared with DFS the binding affinity is lower with DFS. The same trend can be seen when FMSX compared to DFSX. This indicates the isopropyl group on Nitrogen as an important factor which results in variation of binding energies between metabolites of FM. The isopropyl group may be involved in hydrophobic interaction with the binding pocket.

When PH pesticide molecule is converted to its metabolite PC, the ester bond is hydrolyzed to make a carboxylic group, which can make strong H-bonding. As a result, the PC shows higher binding affinity than PH. When comparing metabolite DEMP and PH parent pesticide molecule, ester bond (O connected to P) is hydrolyzed to OH. This also increases the H-bonding ability. This can be seen in results as DEMP has higher binding affinity than PH. DEMPA has a higher H-bonding ability than DEMP due to hydrolyzed ester group. But increment of binding energy was not observed in this case. When DEMPOA is compared with DEMPA, the SH group is substituted with P=O. This transformation increased binding affinity. However, when DEMPO is compared to DEMP, the binding affinity decreased. Therefore, a clear reason for the variation of binding energy between PH and its metabolites is hard to deduce.

When comparing the structure of PF pesticide molecule and its metabolite M1, the P=O oxygen is converted to OH group. This increases the H-bonding ability of M1 compared to PF. This predicted increment of binding energy can be observed in results. O has higher electronegativity than S. Therefore, O-H is more polarized when compared to S-H bond. Therefore, OH group has better H-bonding ability that SH group. Due to this reason, the binding energy of M3 must be lesser than that of M1. This decrement is reflected in results as M3 has less binding affinity than M1. M4 has higher binding affinity than M3. A longer aliphatic chain connected to P via S in M4. This may have increased hydrophobic interaction, resulting in an increase of binding energies. PF and its metabolites have Cl and Br. These groups can make halogen bonding. Cl shows higher halogen bonding as it has the higher electronegativity.

Some forms of monoalkyl and dialkyl phosphates or monoalkyl and dialkyl thiophosphate are metabolites produced in the case of pesticide CP, DM, PH, and CP. When compared with other metabolites and parent pesticide molecules have relatively low binding energies. Only in the case of DM, di and monoalkyl phosphates have high binding energies compared to parent pesticide. This reversal of trend can be due to two reasons. In DM, being a simple molecule compared to other pesticides, the non-phosphate region has very limited ability to make interaction with proteins. The other reason is that DMP and MMP manages to form salt bridges with LYS and ARG. DMP and MMP also make H bonds with ASN, GLU, TRP, and SER.

As mentioned earlier, QP pesticide forms two metabolites: DEPT and HQ. The average binding energy for QP is higher than the HQ metabolite. QP Phosphate derived a part connect to organic ring. This P-SH group can make H-bonds and alkyl groups increase the nonpolar surface for hydrophobic interaction to happen. It is interesting to analyze IM pesticide molecule because most of the above-discussed pesticides are organophosphates, whereas IM is not. When comparing IM with IG metabolite, a decrement of binding affinity can be seen for IM. This may be due to the removal of one H-bonding OH groups.

Overall, binding energy comparison of pesticides and metabolites reveal that, in many cases, metabolites can be more causative towards CKDu compared to their parent pesticides. However, the binding energy cannot dictate the effect of ligand-protein interaction of the biological function of the target proteins. Types of amino acids involved, types of non-covalent interactions, and the location of the binding pocket have to be considered. When discussing about AChE protein, AC, MED, CP, DEP, DETP, DZ, DM, FN, FX, FNS, FXS, FXSX, DFD, DFEX, FM, FMS, FMSX, PH, DEMP, DEMPO, PF, M3, M4, 6-CIPHD, QP, and HQ were bound to the active site of the enzyme, whereas FM, FN, and their metabolites were bound right inside the binding pocket more perfectly than other ligands. These observations reinforce the concept that metabolites can be more potent than the parent pesticides themselves. The binding patterns indicates that TYR341, PHE338, and TRP386 are very important for binding. Further, it is indicated in the literature that organophosphates can inhibit AChE [88]. However, 6-CIPH, which is a metabolite of organochlorine, can in fact bind to the active site as organophosphates do, with roughly similar binding affinity.

When analyzing the results of CP450 interaction with pesticide, a clear inhibition of CP450 cannot be deduced from these results. CP450 can have many allosteric inhibitory sites other than the one discussed above [89]. If any of the locations where ligands bound was such a site, then CP450 can be inhibited. When considering the binding location of ligands to GLS, for pesticides and metabolites, binding site was different than the active site. However, FN, FM, PH, CP and their metabolites and QP bind to a site proximal to the active site. Further, AC manages to bind to the inhibitory site perfectly and IG manages to bind near the inhibitory site. Based on the results, it can be stated that AC has a better chance of hindering the proper function of GLS. FN, FM, PH, CP and their metabolites and also QP can interfere with the binding of substrate to the active site due to the close proximity. It is interesting to see that, even though most of the organophosphates bind to the same place, the secondary interactions formed with amino acid residues are different. This is because the only structural similarity between them is the phosphate group. This emphasizes the importance of phosphate groups for binding.

When considering the GST protein, QP pesticide and its metabolites (HQ and DETP) and also PH pesticide can interfere with the binding of substrate to the active site because of the close proximity. TCP, DFSX, IG, and 6-CIPHD metabolites can also interfere with the binding of substrate to the active site due to the close proximity. These ligands may interfere with the movement of the substrate to the active site. GST protein is inhibited by Ethacrynic Acid (EA) [85]. As IM and AC pesticides manage to bind to the inhibitory site, these pesticides have a chance of hindering the function of GST protein. In the case of ASK1 protein, the active site, ATP binding site, and autophosphorylation site are located in the same pocket. A ligand bound to this pocket can effectively block the active site and ATP binding site or may can as an inhibitor [79]. FN and FM pesticides and their all the metabolites, PH pesticide and its metabolite DEMP, PF pesticide and its metabolite M3, and few other metabolites TCP, IMP, and HQ bind perfectly to this pocket; therefore, these ligands may hinder the proper function of ASK1 by blocking the active site. According to these docking results for the PKC protein, FN and FM pesticides and 6-CIPHD metabolite (metabolite of IM) can block the active site of PKC and can affect the function of PKC catalytic domain.

To obtain a deeper insight into the structural changes, of ligand-protein complexes MD simulations were carried out. Stability of the protein-ligand complexes was confirmed by evaluating RMSD over time, which indicates an equilibrated folded structure in protein-pesticide complexes, as well as protein-metabolite complexes compared to the apo-enzymes. AChE-FMS, AMPK-FNS, ASK1-FMS, and GST–6-CIPHD protein-metabolite complexes have shown a stability similar to their protein-parent pesticide complexes and apo-enzymes. In the case of GST–6-CIPHD protein-metabolite complex, it shows a better stability compared to its protein-parent pesticide complex. Further, the stability of both the protein-pesticide complexes and protein-metabolite complexes was observed through radius of gyration.

Protein residual dynamics was examined by calculating RMSF, and the results indicated that all the proteins considered in MD simulations show similar flexibilities to the apo-enzyme in the presence of metabolites and their relevant pesticide molecules. Further, the RMSF results, together with the analysis of the binding site, revealed that the highly dynamical protein residues for any of the proteins are not involved in ligand interactions. In addition to RMSF, the PCA analysis was carried out to investigate the collective motion of the protein-ligand complexes. According to PCA analysis, AChE and GST protein complexes with metabolites clear changes in conformational motions were observed. Further, PCA analysis for AMPK and ASK1 protein complexes with metabolites showed the presence of comparatively stable clusters in the case of AMPK-FNS and ASK1-FMS protein-metabolite complexes confirming the binding energy results from docking studies and RMSD results from MD studies. Other protein-metabolite complexes considered in MD simulations also showed a similar stability to protein-pesticide complexes. SASA parameter was examined to predict the extent of the conformational changes that occurred during protein-ligand interactions, and according to SASA results, it was observable that all the protein-metabolite complexes considered in MD simulations show similar dynamics as relevant protein-parent pesticide complexes.

Interactions of protein-ligand complexes were further studied using MD simulations in the terms of hydrogen bond analysis, interaction energy calculations, and free energy of binding. The metabolite-protein complexes with more favorable docking binding energies compared to the pesticide-protein complexes showed comparatively higher number of hydrogen bonds, confirming the binding energy results from docking studies and the formation of more stable metabolite-protein complexes according to RMSD results. Further, the interaction energy results and free energy of binding validated the molecular docking results, revealing that pesticide metabolites with higher binding energies according docking studies, effectively bind to the corresponding proteins which are linked to the CKDu, indicating the toxicity of these pesticide metabolites. Therefore, hydrogen bonding analysis, interaction energy results, and free energy of binding, along with the RMSD results, indicated that the metabolites with more favorable binding energies according to docking studies can bind to the corresponding proteins, which are linked to the CKDu effectively and stably, compared to relevant parent pesticide molecules. Thus, in the present study, overall, MD simulation analysis, together with molecular docking studies, reveals that pesticide metabolites, especially M4, FMS, FNS, and 6-CIPHD, have more potential to be toxic towards CKDu compared to parent pesticides. Experimental procedures, such as NMR ligand-detected techniques and X-ray crystallography, to interpret binding modes can further validate the findings of the present study by detecting the pesticide and metabolite binding to renal enzymes.

## 5. Conclusions

According to the molecular docking investigations, it was evident that some pesticides and metabolites could have affinity to bind at the active site or at regulatory sites of studied renal enzymes. In the case of AChE enzyme, AC, CP, DP, DZ, DM, FN, FM, PH, PF, and QP pesticides exhibited much efficient binding at the active site. Metabolites of FM, FN, PF, and IM were predicted to have higher binding energies than that of parent pesticide compounds. When analyzing the docking results of CP450 enzyme, it was found that DZ, PF, and QP bind at the active site. In this case, some metabolites of CP and PF were found to be bound at active sites with considerable binding energies. In the case of GLS enzyme, PH and QP pesticides bound at the active site together with some of their metabolites. PH and QP pesticides, together with metabolites of CP, PH, and IM, bound at the active site of GST. When considering ASK1 enzyme, PH and PF pesticides and metabolites of QP, CP, and PH were found to be bound at the active site. It was evident that, while pesticides FN and FM bind at the active site of PKC, the metabolites of IM also found to be bound at the same site. With regard to AMPK enzyme, only QP pesticide and metabolites of CP were able to bind to the active site. AC pesticide may have some potential to inhibit GLS as it binds with inhibitory site residues. IM and AC might have inhibitory action of GST as it binds at an inhibitory site of GST. These observations positively supported the hypothesis that pesticides can be a possible risk factor towards CKDu as these pesticides and their metabolites bind mainly to active site or regulatory site of considered renal enzymes. This hypothesis was further confirmed through molecular dynamics studies of protein-metabolite and protein-pesticide complexes with higher binding energies. Stability of the protein-ligand complexes was confirmed by evaluating RMSD and over time, and the stable protein-pesticide and protein-metabolite complexes were observed throughout the simulation time. Protein residual dynamics was examined by calculating RMSF, and SASA parameter was examined to predict the extent of the conformational changes that occurred during protein-ligand interactions. According RMSF and SASA results, all the protein-metabolite complexes considered in MD simulations show similar dynamics as relevant protein-parent pesticide complexes. Interactions of protein-ligand complexes were further studied using MD simulations in the terms of hydrogen bond analysis and interaction energy calculations. The protein-metabolite complexes with more favorable binding energies compared to the protein-pesticide complexes showed comparatively higher number of hydrogen bonds and higher interaction energies, confirming the docking results. Therefore, MD simulation analysis, together with molecular docking studies, reveal two important facts: pesticides can be a possible high risk factor towards CKDu; and pesticide metabolites, especially M4, FMS, FNS, and 6-CIPHD, have more potential to be toxic in terms of CKDu compared to parent pesticides.

## Figures and Tables

**Figure 1 biomolecules-11-00261-f001:**
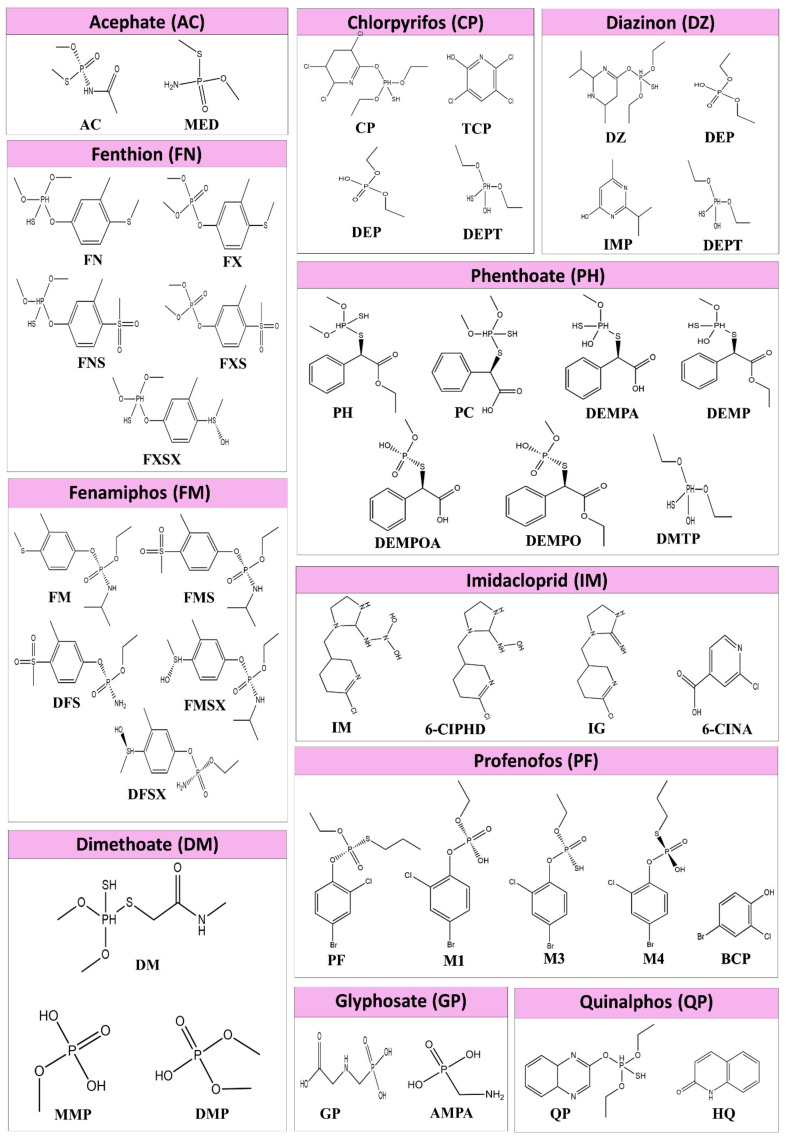
Structures of pesticides and metabolites evaluated in this work. Acephate (AC), chlorpyrifos (CP), diazinon (DZ), dimethoate (DM), fenthion (FN), fenamiphos (FM), phenthoate (PH), profenofos (PF), quinalphos (QP), imidacloprid (IM), and Glyphosate (GP) pesticides and their metabolites, which were experimentally found in urine and and blood samples of poisoned individuals are shown in the figure. The full names for the pesticides and metabolites are mentioned in Appendix A.

**Figure 2 biomolecules-11-00261-f002:**
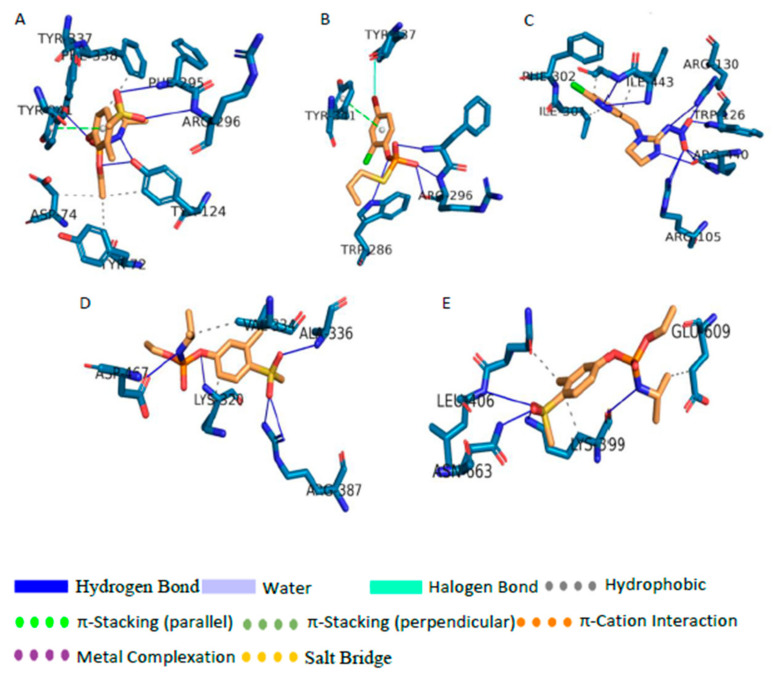
The binding pocket and the non-covalent interactions formed between (**A**) Acetylcholinesterase (AChE) and FMS, (**B**) AChE and M4, (**C**) CP450 and IM, (**D**) Glutaminase (GLS) and FMS, and (**E**) Protein Kinase C (PKC) and FMS.

**Figure 3 biomolecules-11-00261-f003:**
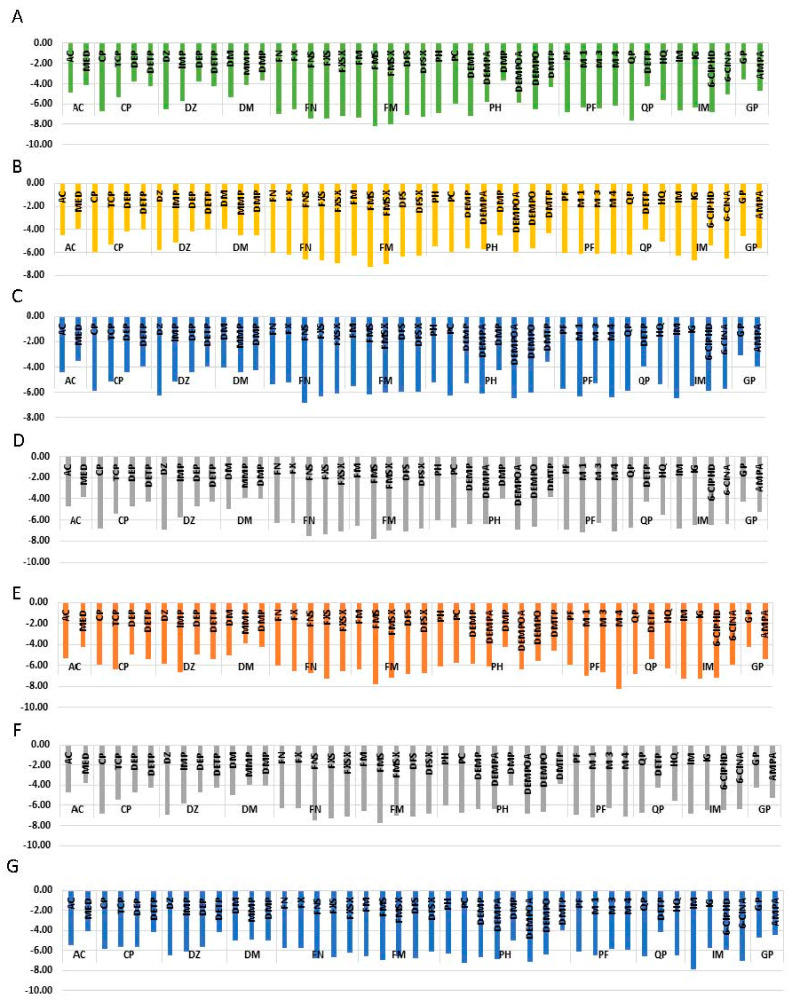
Graphical representation of variation of binding energies kcal/mole (BE) of eight different proteins, i.e., (**A**) AChE, (**B**) Adenosine monophosphate (AMP) Activated Protein Kinase (AMPK), (**C**) Apoptosis Signaling Kinase 1 (ASK1), (**D**) Glutathione S Transferase (GST), (**E**) PKC, (**F**) GLS, and (**G**) CP450 with eleven pesticides and their metabolites.

**Figure 4 biomolecules-11-00261-f004:**
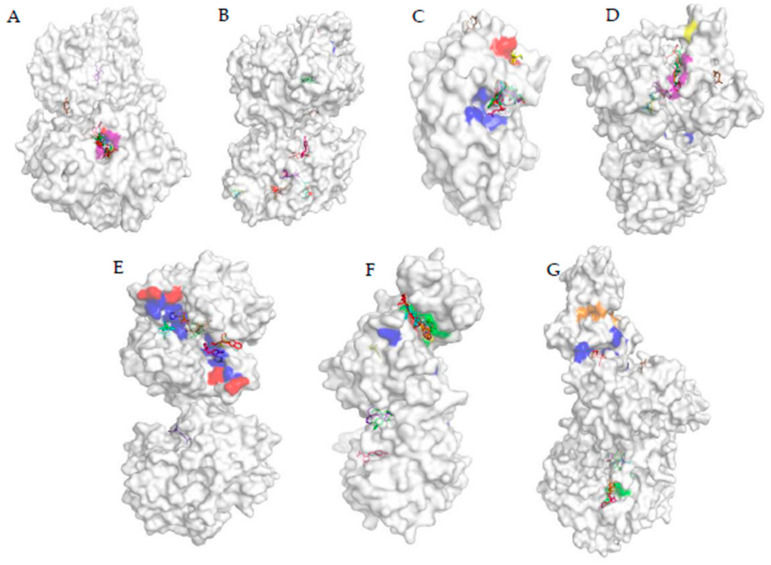
The surface topological view of enzymes where the binding location of ligands are shown relative to important site of enzymes. (**A**) AChE, blue—activity site residues, red—reactivator sit residues, purple—residues that are part of both active and reactivator site. (**B**) CP450, blue—activity site residues, and red—reactivator site residues. (**C**) GLS, blue—activity site residues, red—inhibitor site residues. (**D**) PKC, blue—autophosphorylation site residues, purple—active/inhibitory site residues and yellow—ATP binding site residues. (**E**) GST, blue—activity site residues, red—inhibitor site residues. (**F**) ASK1, yellow—activity site residues, green—ATP binding site, red—inhibition site, and blue—autophosphorylation site. (**G**) AMPK, blue—activity site residues, green—AMP binding site, and orange—activator site.

**Figure 5 biomolecules-11-00261-f005:**
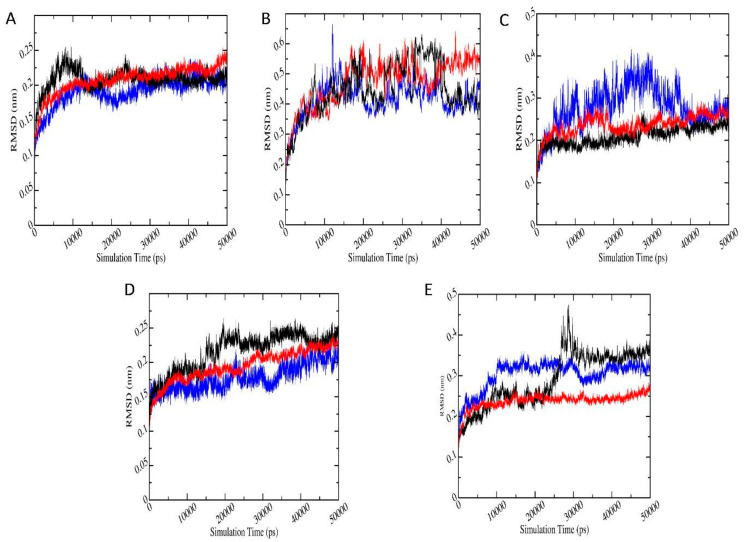
Plot root mean square deviation (RMSD) (nm) versus simulation time (pS), where black, red, and blue color plots represent the parent pesticide, metabolite, and apo-enzyme, respectively. (**A**–**E**) indicate AChE-FM/FMS, AMPK-FN/FNS, ASK1-FM/FMS, GST-IM/6-CIPHD, and PKC-IM/6-CIPHD complexes, respectively.

**Figure 6 biomolecules-11-00261-f006:**
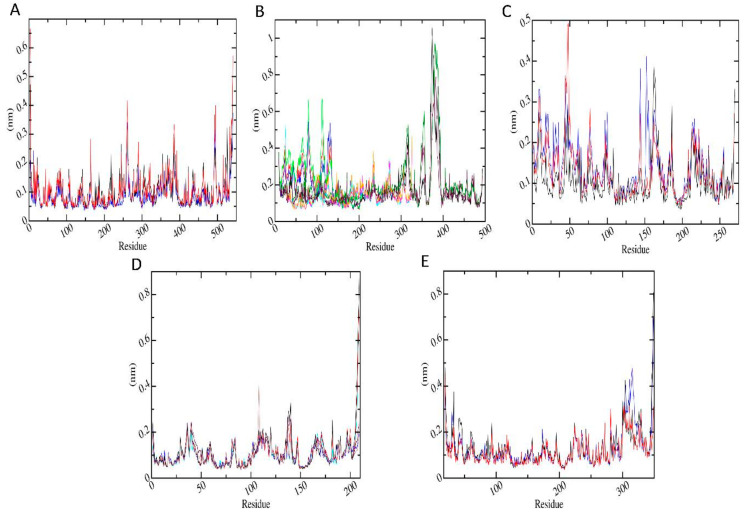
Plot root mean square fluctuation (RMSF) (nm) versus simulation time (pS). Plot **A**, **C**, **E** indicate AChE-FM/FMS, ASK1-FM/FMS, PKC-IM/6-CIPHD complexes, respectively, where black color plots represent parent pesticides, red color plots represent metabolites, and blue color represents apo-enzymes. (**B**) The RMSF of three subunits of AMPK-FN complex are indicated in black, blue, and cyan. Similarly, the RMSF of three subunits of AMPK-FNS complex are indicated in red, magenta, and maroon. Sub-units for apo-AMPK are indicated in dark green, light green, and orange (**D**) The RMSF of two subunits of GST-IM complex are indicated in black and brown. Likewise, the RMSF of two subunits of GST-6CIPHD complex are indicated in red and magenta, while apo-GST subunits are indicated in blue and cyan.

**Figure 7 biomolecules-11-00261-f007:**
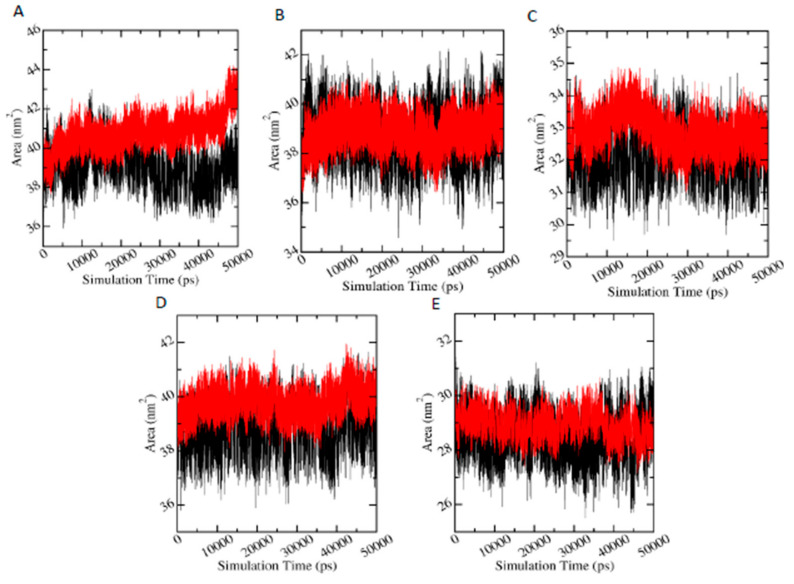
Plot solvent accessible surface area (SASA) (nm^2^) versus simulation time (pS), where black and red color plots represent the parent pesticide and metabolite, respectively. **A**, **B**, **C**, **D**, and **E** indicate AChE-FM/FMS, AMPK-FN/FNS, ASK1-FM/FMS, GST-IM/6-CIPHD, and PKC-IM/6-CIPHD complexes, respectively.

**Figure 8 biomolecules-11-00261-f008:**
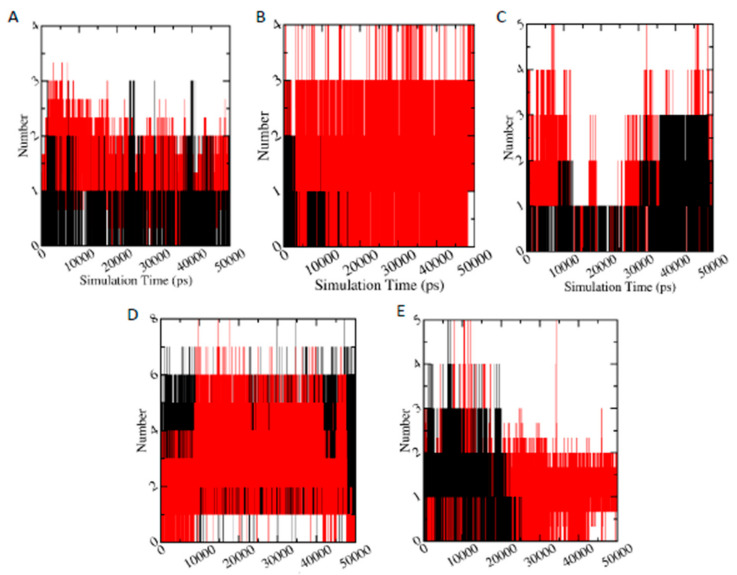
Number of hydrogen bonds versus simulation time (pS), where black and red color plots represent the parent pesticide and metabolite, respectively. (**A**–**E**) indicate AChE-FM/FMS, AMPK-FN/FNS, ASK1-FM/FMS, GST-IM/6-CIPHD, and PKC-IM/6-CIPHD complexes, respectively.

**Figure 9 biomolecules-11-00261-f009:**
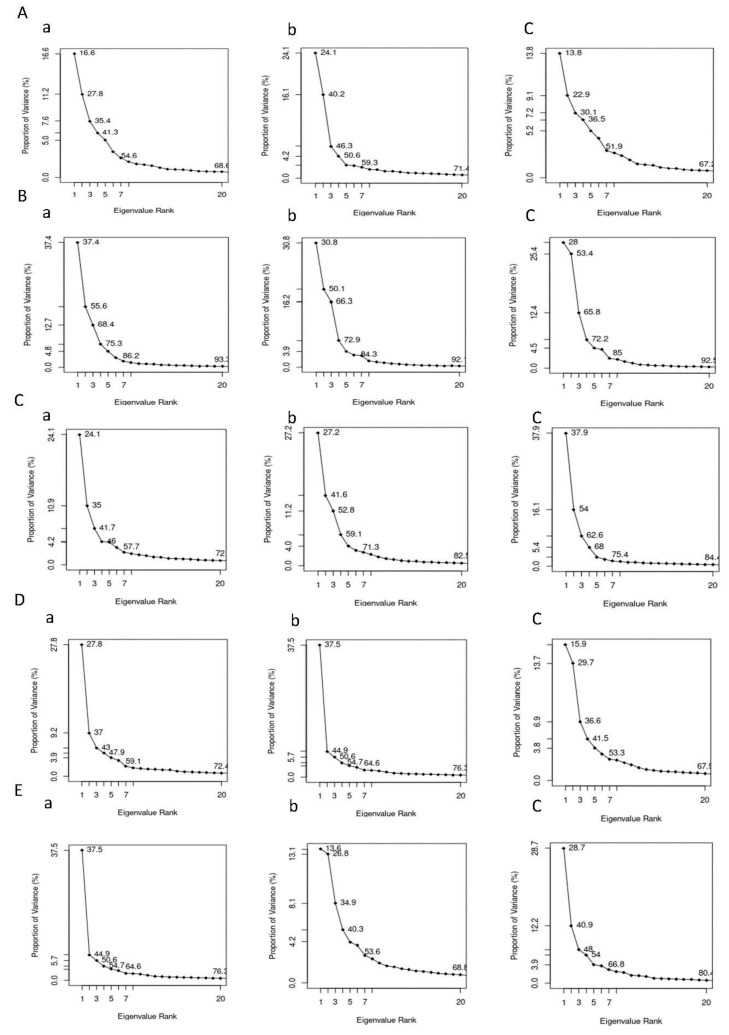
Plot of percentage of variance versus eigenvalue of (**A**) (**a**) AChE-FM, (**b**) AChE-FMS, (**c**) AChE; (**B**) (**a**) AMPK-FN, (**b**) AMPK-FNS, (**c**) AMPK; (**C**) (**a**) ASK1-FM, (**b**) ASK1-FMS, (**c**) ASK1; (**D**) (**a**) GST-IM, (**b**) GST-6-CIPHD, (**c**) GST; and (**E**) (**a**)PKC-IM, (**b**) PKC-6-CIPHD, (**c**) PKC. The color of the principal component analysis (PCA) plot change from blue to red as the complex evolves with time.

**Figure 10 biomolecules-11-00261-f010:**
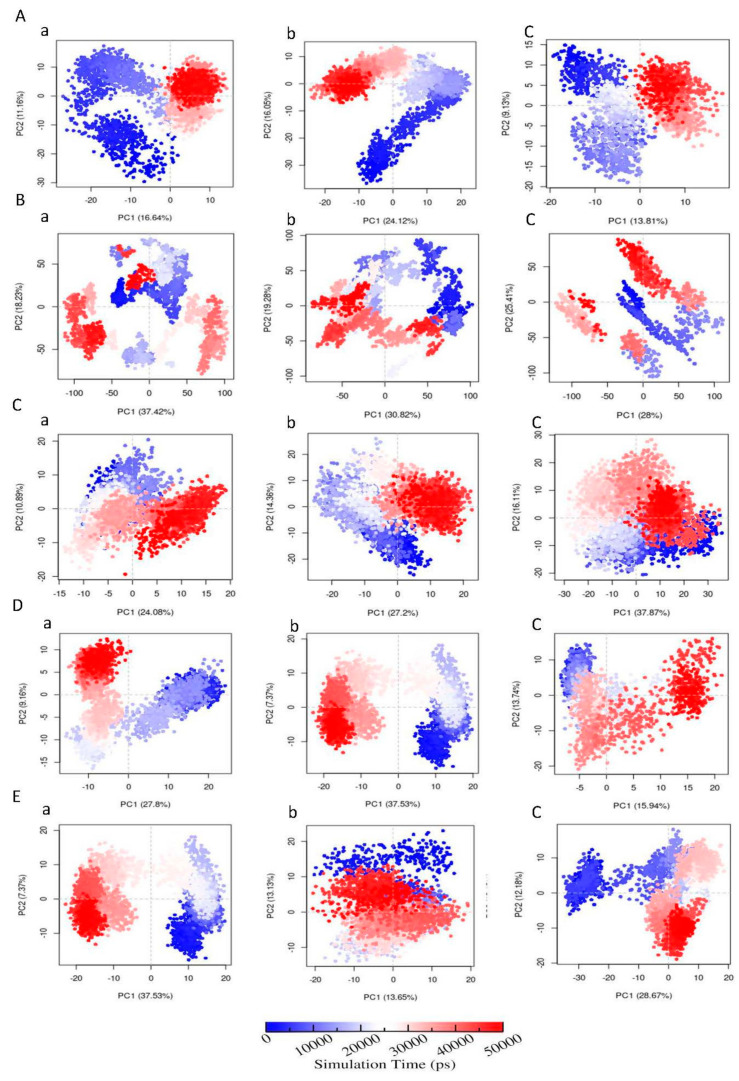
Plot of PC1 versus PC2 of (**A**) (**a**) AChE-FM, (**b**) AChE-FMS, (**c**) AChE; (**B**) (**a**) AMPK-FN, (**b**) AMPK-FNS, (**c**) AMPK; (**C**) (**a**) ASK1-FM, (**b**) ASK1-FMS, (**c**) ASK1; (**D**) (**a**) GST-IM, (**b**) GST-6-CIPHD, (**c**) GST; and (**E**) (**a**) PKC-IM, (**b**) PKC-6-CIPHD, (**c**) PKC. The color of the PCA plot change from blue to red as the complex evolves with time.

**Figure 11 biomolecules-11-00261-f011:**
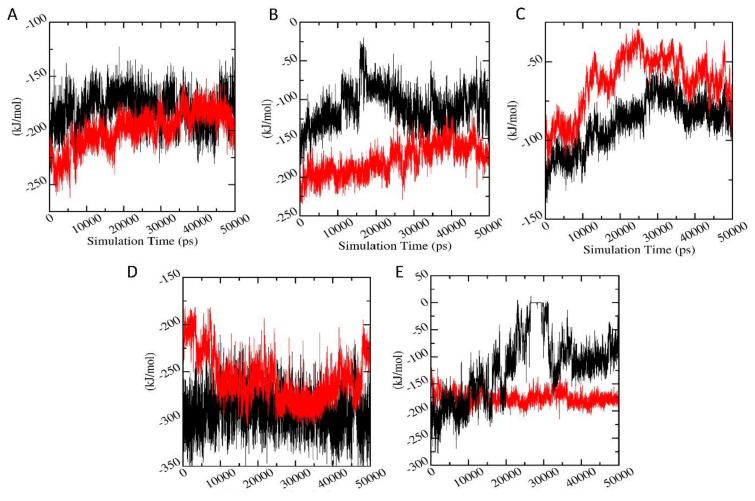
Interaction energy (kJ/mol) versus Simulation time (pS), where black and red color plots represent the parent pesticide and metabolite, respectively. (**A**–**E**) indicate AChE-FM/FMS, AMPK-FN/FNS, ASK1-FM/FMS, GST-IM/6-CIPHD, and PKC-IM/6-CIPHD complexes, respectively.

**Table 1 biomolecules-11-00261-t001:** Protein-pesticide and protein-pesticide metabolite complexes which were considered for the molecular dynamics (MD) simulations.

Protein	Pesticide	Metabolite of the Pesticide
AChE	FM	FMS
AMPK	FN	FNS
ASK1	FM	FMS
GST	IM	6-CIPHD
PKC	IM	6-CIPHD

**Table 2 biomolecules-11-00261-t002:** The comparison between the binding site residues found in literature and theoretical binding residues found by docking.

Protein	Chemical Identity	Action	Binding Residues From Literature	Binding Energy (kcal/mol)	Theoretical Binding Residues
AMPK	5-{[6-chloro-5-(2′-hydroxy[1,1′-biphenyl]-4-yl)-1H-benzimidazol-2-yl]oxy}-N-hydroxy-2-methylbenzamide	Activator	ARG83B, ASN50A, VAL113B, ASP108B, GLY30A, LY33A, ILE48A, VAL13A, LEU20A, VAL81B [46]	−9.52	ARG83B, ASN50A, VAL113B, ASP108B, GLY30A, LY33A, ILE48A, VAL13A, LEU20A,
PKC	3-{1-[3-(dimethylamino)propyl]-2-methyl-1h-indol-3-yl}-4-(2-methyl-1h-indol-3-yl)-1h-pyrrole-2,5-dione	Inhibitor	GLU421A, THR404A, VAL356A, LEU348A, ASP484A, ALA483A, PHE353A [47]	−8.42	GLU421A, THR404A, VAL356A, LEU348A, ASP484A, ALA483A, PHE353A
GLS	5,5′-(sulfanediyldiethane-2,1-diyl)bis(1,3,4-thiadiazol-2-amine)	Inhibitor	LEU323A, TYR394A [48]	−9.37	LEU323A, TYR394A
ASK1	4-tert-butyl-N-[6-(1H-imidazol-1-yl)imidazo[1,2-a]pyridin-2-yl]benzamide	Inhibitor	LYS709A, PRO758A, VAL649A, LEU810A, ALA707A, VAL757A [49]	−8.25	LYS709A, PRO758A, VAL649A, ALA707A, VAL757A
AChE	1-[({2,4-bis[(e)-(hydroxyimino)methyl]pyridinium-1-yl}methoxy)methyl]-4-carbamoylpyridinium	Reactivator	TYR337A, PHE338A, TYR341A, TRP286A, VAL282A, ASP74A, SER125A, ASN87A, TYR72A, TYR124A [50]	−7.98	TYR337A, PHE338A, TRP286A, VAL282A, SER125A, ASN87A, TYR72A, TYR124A

**Table 3 biomolecules-11-00261-t003:** Radius of gyration of the crystal, protein-ligand complex, and protein during the MD simulation.

Protein	Rg of Crystal Structure of Protein (nm)Metabolite	Average Rg and Standard Deviation of Protein Ligand Complex During MD Simulation		Average Rg and Standard Deviation of Protein during MD Simulation	
		Metabolite	Parent Pesticide	Metabolite	Parent Pesticide
AChE	2.266	2.308 (±0.041)	2.308 (±0.083)	2.312 (±0.001)	2.312 (±0.001)
ASK1	1.901	1.957 (±0.013)	1.918 (±0.008)	1.939 (±0.001)	1.924 (±0.001)
PKC	2.0541	2.075 (±0.006)	2.081 (±0.001)	2.079 (±0.019)	2.082 (±0.013)
GST	2.104	2.127 (±0.005)	2.113 (±0.007)	2.132 (±0.017)	2.117 (±0.067)
AMPK	3.427	3.452 (±0.016)	3.478 (±0.020)	3.452 (±0.022)	3.475 (±0.022)

**Table 4 biomolecules-11-00261-t004:** The binding free energy summation of the polar, non-polar energies, and non-bonded interaction energies (Vander Waals and electrostatic interaction) and interaction entropy for protein-ligand complexes.

Complex	VdE Energy (kJ/mol)	Elec. Energy (kJ/mol)	Polar Solvation Energy (kJ/mol)	SASA Energy (kJ/mol)	Binding Energy (kJ/mol)	Entropy of Binding TΔS (kJ/mol)
AChE-FM	−180.89 ± 8.92	−16.43 ± 6.58	115.53 ± 10.37	−18.77 ± 0.65	−100.56 ± 12.50	−14.36
AChE-FMS	−175.27 ± 9.22	−13.34 ± 6.56	94.07 ± 18.03	−18.81 ± 1.04	−113.34 ± 15.42	−15.84
ASK1-FM	−116.32 ± 9.10	−35.00 ± 6.67	92.26 ± 7.07	−14.44 ± 0.74	−73.49 ± 9.24	−11.44
ASK1-FMS	−116.61 ± 20.747	−8.22 ± 1.265	78.60 ± 17.334	−14.00 ± 2.015	−60.23 ± 14.03	−31.2
AMPK-FN	−95.75 ± 14.32	−13.72 ± 9.25	58.96 ± 13.21	10.70 ± 0.64	−61.20 ± 13.31	−15.59
AMPK-FNS	−118.99 ± 6.80	−21.62 ± 1.56	83.24 ± 20.22	−13.35 ± 0.73	−70.72 ± 14.7	−25.30
PKC-IM	−77.04 ± 12.46	−29.36 ± 3.61	75.44 ± 13.32	−8.679 ± 1.0	−39.64 ± 15.40	−42.809
PKC-6CIPHD	−109.58 ± 12.92	−21.77 ± 9.54	91.09 ± 26.82	−12.78 ± 1.15	−53.03 ± 24.60	−14.22
GST-IM	−108.88 ± 9.38	−113.57 ± 12.44	254.65 ± 17.67	−13.27 ± 0.51	18.92 ± 7.71	−17.449
GST-6CIPHD	−137.52 ± 6.45	−137.52 ± 6.45	231.87 ± 23.23	−13.83 ± 0.73	−21.40 ± 10.36	−17.443

## Data Availability

Not applicable.

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
