# Peer review of "Demystifying Chronic Kidney Disease of Unknown Etiology (CKDu): Computational Interaction Analysis of Pesticides and Metabolites with Vital Renal Enzymes"

_biomolecules, 2021, doi:10.3390/biom11020261_

Round 1

Reviewer 1 Report

See doc in attach

Reviewer 2 Report

The work "Demystifying Chronic Kidney Disease of Unknown etiology (CKDu): Computational Interaction Analysis of Pesticides and Metabolites with Vital Renal Enzymes" is a tour-de-force of docking and molecular simulations that seek to ask to what extent could common pesticides inhibit enzymes necessary for kidney function. The work is very interesting and the approach quite unique (the reviewer is aware of only one other large-scale effort to examine toxic effects from molecular docking point-of-view, see: https://journals.plos.org/plosone/article?id=10.1371/journal.pone.0106298).

That being said, I do have a few suggestions/comments for the authors:

"All protein-ligand complexes show stability in 50ns simulation" This is not quite accurate: systems corresponding to D and E seem to show significant drifts and B has a fairly high RMSD. It could be argued that these systems have adopted an alternative state relative to the reference state and are 'stable' near this state. Generating a per-frame RMSD matrix (frame-vs-frame RMSD) map would be of interest when addressing this stability question.

The work on PKD2 seems incomplete for this work, and I would advise excluding it from this manuscript. It is certainly of interest to understand how different ligands interact with PKD2 and it's channel function; however, given the amount work already presented for the other target-proteins, it seems premature to include a discussion of that protein here.

A comment on the figure resolution: It is difficult to read the labels on many of the figures. Perhaps the authors can re-render them or bold the text to make the axis a bit more legible.

It is unclear what the value of the PCA analysis is here.

One set of calculations that seems absent from the analysis presented here are apo-state simulations. A comparison of the parent vs metabolite is quite interesting; however, it would be interesting if the authors could compare to a 'common ground' when discussing protein motions (such as the PCA work). The PCA on its own doesn't seem to add much, but if the ligand-bound systems were projected onto the apo-state principle components and the authors discussed what changes in motions have occurred (or not occurred) for the parent vs metabolites relative to the apo-state, that would be quite interesting.

Interaction energies on their own exclude the entropic component of the free-energy of binding. It would be interesting if the authors instead of showing interaction energy time-series, provided instead an estimate of the differences in binding free-energies of the parent and metabolite ligands. A quick way to do this would be to compute average interaction energies and the interaction entropies (see: https://pubs.acs.org/doi/10.1021/jacs.6b02682 ).

Overall, the work is very interesting.

Round 2

Reviewer 1 Report

All the questions have been answered by the authors.

Reviewer 2 Report

The author's have addressed my concerns.